# CSI: Novelty Detection via Contrastive Learning on Distributionally Shifted Instances

**Jihoon Tack**[*†], **Sangwoo Mo**[*‡], **Jongheon Jeong**[‡], **Jinwoo Shin**[†‡]
[†]Graduate School of AI, KAIST
[‡]School of Electrical Engineering, KAIST
{jihoontack,swmo,jongheonj,jinwoos}@kaist.ac.kr

## Abstract

Novelty detection, *i.e.*, identifying whether a given sample is drawn from outside the training distribution, is essential for reliable machine learning. To this end, there have been many attempts at learning a representation well-suited for novelty detection and designing a score based on such representation. In this paper, we propose a simple, yet effective method named *contrasting shifted instances* (CSI), inspired by the recent success on contrastive learning of visual representations. Specifically, in addition to contrasting a given sample with other instances as in conventional contrastive learning methods, our training scheme contrasts the sample with distributionally-shifted augmentations of itself. Based on this, we propose a new detection score that is specific to the proposed training scheme. Our experiments demonstrate the superiority of our method under various novelty detection scenarios, including unlabeled one-class, unlabeled multi-class and labeled multi-class settings, with various image benchmark datasets. Code and pre-trained models are available at https://github.com/alinlab/CSI.

## 1 Introduction

Out-of-distribution (OOD) detection [26], also referred to as a novelty- or anomaly detection is a task of identifying whether a test input is drawn far from the training distribution (in-distribution) or not. In general, the OOD detection problem aims to detect OOD samples where a detector is allowed to access only to training data. The space of OOD samples is typically huge, *i.e.*, an OOD sample can vary significantly and arbitrarily from the given training distribution. Hence, assuming specific prior knowledge, *e.g.*, external data representing some specific OODs, may introduce a bias to the detector. The OOD detection is a classic yet essential problem in machine learning, with a broad range of applications, including medical diagnosis [4], fraud detection [53], and autonomous driving [12].

A long line of literature has thus been proposed, including density-based [74, 46, 6, 47, 11, 55, 61, 17], reconstruction-based [58, 76, 9, 54, 52, 7], one-class classifier [59, 56], and self-supervised [15, 25, 2] approaches. Overall, a majority of recent literature is concerned with (a) modeling the representation to better encode normality [23, 25], and (b) defining a new detection score [56, 2]. In particular, recent studies have shown that inductive biases from self-supervised learning significantly help to learn discriminative features for OOD detection [15, 25, 2].

Meanwhile, recent progress on self-supervised learning has proven the effectiveness of *contrastive learning* in various domains, *e.g.*, computer vision [21, 5], audio processing [50], and reinforcement learning [63]. Contrastive learning extracts a strong inductive bias from multiple (similar) views of a sample by let them attract each other, yet repelling them to other samples. *Instance discrimination* [69]

---

[*]Equal contribution

is a special type of contrastive learning where the views are restricted up to different augmentations, which have achieved state-of-the-art results on visual representation learning [21, 5].

Inspired by the recent success of instance discrimination, we aim to utilize its power of representation learning for OOD detection. To this end, we investigate the following questions: (a) how to learn a (more) discriminative representation for detecting OODs and (b) how to design a score function utilizing the representation from (a). We remark that the desired representation for OOD detection may differ from that for standard representation learning [23, 25], as the former aims to discriminate in-distribution and OOD samples, while the latter aims to discriminate *within* in-distribution samples.

We first found that existing contrastive learning scheme is already reasonably effective for detecting OOD samples with a proper detection score. We further observe that one can improve its performance by utilizing "hard" augmentations, *e.g.*, rotation, that were known to be harmful and unused for the standard contrastive learning [5]. In particular, while the existing contrastive learning schemes act by pulling all augmented samples toward the original sample, we suggest to additionally push the samples with hard or distribution-shifting augmentations away from the original. We observe that contrasting shifted samples help OOD detection, as the model now learns a new task of discriminating *between* in- and out-of-distribution, in addition to the original task of discriminating *within* in-distribution.

**Contribution.** We propose a simple yet effective method for OOD detection, coined *contrasting shifted instances* (CSI). Built upon the existing contrastive learning scheme [5], we propose two novel additional components: (a) a new training method which contrasts distributionally-shifted augmentations (of the given sample) in addition to other instances, and (b) a score function which utilizes both the contrastively learned representation and our new training scheme in (a). Finally, we show that CSI enjoys broader usage by applying it to improve the confidence-calibration of the classifiers: it relaxes the overconfidence issue in their predictions for both in- and out-of-distribution samples while maintaining the classification accuracy.

We verify the effectiveness of CSI under various environments of detecting OOD, including unlabeled one-class, unlabeled multi-class, and labeled multi-class settings. To our best knowledge, we are the first to demonstrate all three settings under a single framework. Overall, CSI outperforms the baseline methods for all tested datasets. In particular, CSI achieves new state-of-the-art results[2] on one-class classification, *e.g.*, it improves the mean area under the receiver operating characteristics (AUROC) from 90.1% to 94.3% (+4.2%) for CIFAR-10 [33], 79.8% to 89.6% (+9.8%) for CIFAR-100 [33], and 85.7% to 91.6% (+5.9%) for ImageNet-30 [25] one-class datasets, respectively. We remark that CSI gives a larger improvement in harder (or near-distribution) OOD samples. To verify this, we also release new benchmark datasets: fixed version of the resized LSUN and ImageNet [39].

We remark that learning representation to discriminate in- vs. out-of-distributions is an important but under-explored problem. We believe that our work would guide new interesting directions in the future, for both representation learning and OOD detection.

## 2 CSI: Contrasting shifted instances

For a given dataset $\{x_m\}_{m=1}^M$ sampled from a data distribution $p_{\text{data}}(x)$ on the data space $\mathcal{X}$, the goal of out-of-distribution (OOD) detection is to model a detector from $\{x_m\}$ that identifies whether $x$ is sampled from the data generating distribution (or in-distribution) $p_{\text{data}}(x)$ or not. As modeling $p_{\text{data}}(x)$ directly is prohibitive in most cases, many existing methods for OOD detection define a *score function* $s(x)$ that a high value heuristically represents that $x$ is from in-distribution.

### 2.1 Contrastive learning

The idea of *contrastive learning* is to learn an encoder $f_\theta$ to extract the necessary information to distinguish similar samples from the others. Let $x$ be a query, $\{x_+\}$, and $\{x_-\}$ be a set of positive and negative samples, respectively, and $\text{sim}(z, z') := z \cdot z'/\|z\|\|z'\|$ be the cosine similarity. Then, the primitive form of the *contrastive loss* is defined as follows:

$$\mathcal{L}_{\text{con}}(x, \{x_+\}, \{x_-\}) := -\frac{1}{|\{x_+\}|} \log \frac{\sum_{x' \in \{x_+\}} \exp(\text{sim}(z(x), z(x'))/\tau)}{\sum_{x' \in \{x_+\} \cup \{x_-\}} \exp(\text{sim}(z(x), z(x'))/\tau)}, \quad (1)$$

where $|\{x_+\}|$ denotes the cardinality of the set $\{x_+\}$, $z(x)$ denotes the output feature of the contrastive layer, and $\tau$ denotes a temperature hyper-parameter. One can define the contrastive feature $z(x)$

directly from the encoder $f_\theta$, *i.e.*, $z(x) = f_\theta(x)$ [21], or apply an additional projection layer $g_\phi$, *i.e.*, $z(x) = g_\phi(f_\theta(x))$ [5]. We use the projection layer following the recent studies [5, 30].

In this paper, we specifically consider the simple contrastive learning (*SimCLR*) [5], a simple and effective objective based on the task of *instance discrimination* [69]. Let $\tilde{x}_i^{(1)}$ and $\tilde{x}_i^{(2)}$ be two independent augmentations of $x_i$ from a pre-defined family $\mathcal{T}$, namely, $\tilde{x}^{(1)} := T_1(x_i)$ and $\tilde{x}^{(2)} := T_2(x_i)$, where $T_1, T_2 \sim \mathcal{T}$. Then the SimCLR objective can be defined by the contrastive loss (1) where each $(\tilde{x}_i^{(1)}, \tilde{x}_i^{(2)})$ and $(\tilde{x}_i^{(2)}, \tilde{x}_i^{(1)})$ are considered as query-key pairs while others being negatives. Namely, for a given batch $\mathcal{B} := \{x_i\}_{i=1}^B$, the SimCLR objective is defined as follows:

$$\mathcal{L}_{\texttt{SimCLR}}(\mathcal{B}; \mathcal{T}) := \frac{1}{2B} \sum_{i=1}^B \mathcal{L}_{\texttt{con}}(\tilde{x}_i^{(1)}, \tilde{x}_i^{(2)}, \tilde{\mathcal{B}}_{-i}) + \mathcal{L}_{\texttt{con}}(\tilde{x}_i^{(2)}, \tilde{x}_i^{(1)}, \tilde{\mathcal{B}}_{-i}), \tag{2}$$

where $\tilde{\mathcal{B}} := \{\tilde{x}_i^{(1)}\}_{i=1}^B \cup \{\tilde{x}_i^{(2)}\}_{i=1}^B$ and $\tilde{\mathcal{B}}_{-i} := \{\tilde{x}_j^{(1)}\}_{j \neq i} \cup \{\tilde{x}_j^{(2)}\}_{j \neq i}$.

## 2.2 Contrastive learning for distribution-shifting transformations

Chen et al. [5] has performed an extensive study on which family of augmentations $\mathcal{T}$ leads to a better representation when used in SimCLR, *i.e.*, which transformations should $f_\theta$ consider as positives. Overall, the authors report that some of the examined augmentations (*e.g.*, rotation), sometimes degrades the discriminative performance of SimCLR. One of our key findings is that such augmentations can be useful for OOD detection by considering them as *negatives* - contrast from the original sample. In this paper, we explore which family of augmentations $\mathcal{S}$, which we call *distribution-shifting transformations*, or simply *shifting transformations*, would lead to better representation in terms of OOD detection when used as negatives in SimCLR.

**Contrasting shifted instances.** We consider a set $\mathcal{S}$ consisting of $K$ different (random or deterministic) transformations, including the identity $I$: namely, we denote $\mathcal{S} := \{S_0 = I, S_1, \ldots, S_{K-1}\}$. In contrast to the vanilla SimCLR that considers augmented samples as positive to each other, we attempt to consider them as negative if the augmentation is from $\mathcal{S}$. For a given batch of samples $\mathcal{B} = \{x_i\}_{i=1}^B$, this can be done simply by augmenting $\mathcal{B}$ via $\mathcal{S}$ before putting it into the SimCLR loss defined in (2): namely, we define *contrasting shifted instances* (con-SI) loss as follows:

$$\mathcal{L}_{\texttt{con-SI}} := \mathcal{L}_{\texttt{SimCLR}}\left( \bigcup_{S \in \mathcal{S}} \mathcal{B}_S; \mathcal{T} \right), \quad \text{where } \mathcal{B}_S := \{S(x_i)\}_{i=1}^B. \tag{3}$$

Here, our intuition is to regard each distributionally-shifted sample (*i.e.*, $S \neq I$) as an OOD with respect to the original. In this respect, con-SI attempts to discriminate an in-distribution (*i.e.*, $S = I$) sample from other OOD (*i.e.*, $S \in \{S_1, \ldots, S_{K-1}\}$) samples. We further verify the effectiveness of con-SI in our experimental results: although con-SI does not improve representation for standard classification, it does improve OOD detection significantly (see linear evaluation in Section 3.2).

**Classifying shifted instances.** In addition to contrasting shifted instances, we consider an auxiliary task that predicts which shifting transformation $y^S \in \mathcal{S}$ is applied for a given input $x$, in order to facilitate $f_\theta$ to discriminate each shifted instance. Specifically, we add a linear layer to $f_\theta$ for modeling an auxiliary softmax classifier $p_{\texttt{cls-SI}}(y^\mathcal{S}|x)$, as in [15, 25, 2]. Let $\tilde{\mathcal{B}}_S$ be the batch augmented from $\mathcal{B}_S$ via SimCLR; then, we define *classifying shifted instances* (cls-SI) loss as follows:

$$\mathcal{L}_{\texttt{cls-SI}} := \frac{1}{2B} \frac{1}{K} \sum_{S \in \mathcal{S}} \sum_{\tilde{x}_S \in \tilde{\mathcal{B}}_S} -\log p_{\texttt{cls-SI}}(y^\mathcal{S} = S \mid \tilde{x}_S). \tag{4}$$

The final loss of our proposed method, *CSI*, is defined by combining the two objectives:

$$\mathcal{L}_{\texttt{CSI}} = \mathcal{L}_{\texttt{con-SI}} + \lambda \cdot \mathcal{L}_{\texttt{cls-SI}} \tag{5}$$

where $\lambda > 0$ is a balancing hyper-parameter. We simply set $\lambda = 1$ for all our experiments.

***OOD-ness*: How to choose the shifting transformation?** In principle, we choose the shifting transformation that generates the most OOD-like yet semantically meaningful samples. Intuitively, such samples can be most effective ('nearby' but 'not-too-nearby') OOD samples, as also discussed in Section 3.2. More specifically, we measure the *OOD-ness* of a transformation by the area under the receiver operating characteristics (AUROC) between in-distribution vs. transformed samples under vanilla SimCLR, using the detection score (6) defined in Section 2.3. The transformation with high OOD-ness values (*i.e.*, OOD-like) indeed performs better (see Table 4 and Table 5 in Section 3.2).

## 2.3 Score functions for detecting out-of-distribution

Upon the representation $z(\cdot)$ learned by our proposed training objective, we define several score functions for detecting out-of-distribution; whether a given $x$ is OOD or not. We first propose a detection score that is applicable to any contrastive representation. We then introduce how one could incorporate additional information learned by contrasting (and classifying) shifted instances as in (5).

**Detection score for contrastive representation.** Overall, we find that two features from SimCLR representations are surprisingly effective for detecting OOD samples: (a) the *cosine similarity* to the nearest training sample in $\{x_m\}$, *i.e.*, $\max_m \mathrm{sim}(z(x_m), z(x))$, and (b) the *norm* of the representation, *i.e.*, $\|z(x)\|$. Intuitively, the contrastive loss increases the norm of in-distribution samples, as it is an easy way to minimize the cosine similarity of identical samples by increasing the denominator of (1). We discuss further detailed analysis of both features in Appendix H. We simply combine these features to define a detection score $s_{\mathrm{con}}$ for contrastive representation:

$$s_{\mathrm{con}}(x; \{x_m\}) := \max_m \ \mathrm{sim}(z(x_m), z(x)) \cdot \|z(x)\|. \tag{6}$$

We also discuss how one can reduce the computation and memory cost by choosing a proper subset (*i.e.*, coreset) of training samples in Appendix E.

**Utilizing shifting transformations.** Given that our proposed $\mathcal{L}_{\mathrm{CSI}}$ is used for training, one can further improve the detection score $s_{\mathrm{con}}$ significantly by incorporating shifting transformations $\mathcal{S}$. Here, we propose two additional scores, $s_{\mathrm{con\text{-}SI}}$ and $s_{\mathrm{cls\text{-}SI}}$, where are corresponded to $\mathcal{L}_{\mathrm{con\text{-}SI}}$ (3) and $\mathcal{L}_{\mathrm{cls\text{-}SI}}$ (4), respectively.

Firstly, we define $s_{\mathrm{con\text{-}SI}}$ by taking an expectation of $s_{\mathrm{con}}$ over $S \in \mathcal{S}$:

$$s_{\mathrm{con\text{-}SI}}(x; \{x_m\}) := \sum_{S \in \mathcal{S}} \lambda_S^{\mathrm{con}} \ s_{\mathrm{con}}(S(x); \{S(x_m)\}), \tag{7}$$

where $\lambda_S^{\mathrm{con}} := M / \sum_m s_{\mathrm{con}}(S(x_m); \{S(x_m)\}) = M / \sum_m \|z(S(x_m))\|$ for $M$ training samples is a balancing term to scale the scores of each shifting transformation (See Appendix F for details).

Secondly, we define $s_{\mathrm{cls\text{-}SI}}$ utilizing the auxiliary classifier $p(y^{\mathcal{S}}|x)$ upon $f_\theta$ as follows:

$$s_{\mathrm{cls\text{-}SI}}(x) := \sum_{S \in \mathcal{S}} \lambda_S^{\mathrm{cls}} \ W_S f_\theta(S(x)), \tag{8}$$

where $\lambda_S^{\mathrm{cls}} := M / \sum_m [W_S f_\theta(S(x_m))]$ are again balancing terms similarly to above, and $W_S$ is the weight vector in the linear layer of $p(y^{\mathcal{S}}|x)$ per $S \in \mathcal{S}$.

Finally, the combined score for CSI representation is defined as follows:

$$s_{\mathrm{CSI}}(x; \{x_m\}) := s_{\mathrm{con\text{-}SI}}(x; \{x_m\}) + s_{\mathrm{cls\text{-}SI}}(x). \tag{9}$$

**Ensembling over random augmentations.** In addition, we find one can further improve each of the proposed scores by ensembling it over random augmentations $T(x)$ where $T \sim \mathcal{T}$. Namely, for instance, the *ensembled* CSI score is defined by $s_{\mathrm{CSI\text{-}ens}}(x) := \mathbb{E}_{T \sim \mathcal{T}}[s_{\mathrm{CSI}}(T(x))]$. Unless otherwise noted, we use these ensembled versions of (6) to (9) in our experiments. See Appendix D for details.

## 2.4 Extension for training confidence-calibrated classifiers

Furthermore, we propose an extension of CSI for training *confidence-calibrated* classifiers [22, 37] from a given labeled dataset $\{(x_m, y_m)\}_m \subseteq \mathcal{X} \times \mathcal{Y}$ by adapting it to *supervised contrastive learning* (SupCLR) [30]. Here, the goal is to model a classifier $p(y|x)$ that is (a) accurate on predicting $y$ when $x$ is in-distribution, and (b) the *confidence* $s_{\mathrm{sup}}(x) := \max_y p(y|x)$ [22] of the classifier is *well-calibrated*, *i.e.*, $s_{\mathrm{sup}}(x)$ should be low if $x$ is an OOD sample or $\arg\max_y p(y|x) \neq$ true label.

**Supervised contrastive learning (SupCLR).** SupCLR is a supervised extension of SimCLR that contrasts samples in *class-wise*, instead of in instance-wise: every samples of the same classes are considered as positives. Let $\mathcal{C} = \{(x_i, y_i)\}_{i=1}^B$ be a training batch with class labels $y_i \in \mathcal{Y}$, and $\tilde{\mathcal{C}}$ be an augmented batch by random transformation $\mathcal{T}$, *i.e.*, $\tilde{\mathcal{C}} := \{(\tilde{x}_j, y_j) \mid \tilde{x}_j \in \tilde{\mathcal{B}}\}$. For a given label $y$, we divide $\tilde{\mathcal{C}}$ into two subsets $\tilde{\mathcal{C}} = \tilde{\mathcal{C}}_y \cup \tilde{\mathcal{C}}_{-y}$ where $\tilde{\mathcal{C}}_y$ contains the samples of label $y$ and $\tilde{\mathcal{C}}_{-y}$ contains the remaining. Then, the SupCLR objective is defined by:

$$\mathcal{L}_{\mathrm{SupCLR}}(\mathcal{C}; \mathcal{T}) := \frac{1}{2B} \sum_{j=1}^{2B} \mathcal{L}_{\mathrm{con}}(\tilde{x}_j, \tilde{\mathcal{C}}_{y_j} \setminus \{\tilde{x}_j\}, \tilde{\mathcal{C}}_{-y_j}). \tag{10}$$

Table 1: AUROC (%) of various OOD detection methods trained on one-class dataset of (a) CIFAR-10, (b) CIFAR-100 (super-class), and (c) ImageNet-30. For CIFAR-10, we report the means and standard deviations of per-class AUROC averaged over five trials, and the final column indicates the mean AUROC across all the classes. For CIFAR-100 and ImaegeNet-30, we only report the mean AUROC over a single trial. Bold denotes the best results, and $^*$ denotes the values from the reference. See Appendix C for additional results, *e.g.*, per-class AUROC on CIFAR-100 and ImageNet-30.

(a) One-class CIFAR-10

| Method | Network | Plane | Car | Bird | Cat | Deer | Dog | Frog | Horse | Ship | Truck | Mean |
|---|---|---|---|---|---|---|---|---|---|---|---|---|
| OC-SVM$^*$ [59] | - | 65.6 | 40.9 | 65.3 | 50.1 | 75.2 | 51.2 | 71.8 | 51.2 | 67.9 | 48.5 | 58.8 |
| DeepSVDD$^*$ [56] | LeNet | 61.7 | 65.9 | 50.8 | 59.1 | 60.9 | 65.7 | 67.7 | 67.3 | 75.9 | 73.1 | 64.8 |
| AnoGAN$^*$ [58] | DCGAN | 67.1 | 54.7 | 52.9 | 54.5 | 65.1 | 60.3 | 58.5 | 62.5 | 75.8 | 66.5 | 61.8 |
| OCGAN$^*$ [52] | OCGAN | 75.7 | 53.1 | 64.0 | 62.0 | 72.3 | 62.0 | 72.3 | 57.5 | 82.0 | 55.4 | 65.7 |
| Geom$^*$ [15] | WRN-16-8 | 74.7 | 95.7 | 78.1 | 72.4 | 87.8 | 87.8 | 83.4 | 95.5 | 93.3 | 91.3 | 86.0 |
| Rot$^*$ [25] | WRN-16-4 | 71.9 | 94.5 | 78.4 | 70.0 | 77.2 | 86.6 | 81.6 | 93.7 | 90.7 | 88.8 | 83.3 |
| Rot+Trans$^*$ [25] | WRN-16-4 | 77.5 | 96.9 | 87.3 | 80.9 | 92.7 | 90.2 | 90.9 | 96.5 | 95.2 | 93.3 | 90.1 |
| GOAD$^*$ [2] | WRN-10-4 | 77.2 | 96.7 | 83.3 | 77.7 | 87.8 | 87.8 | 90.0 | 96.1 | 93.8 | 92.0 | 88.2 |
| Rot [25] | ResNet-18 | 78.3±0.2 | 94.3±0.3 | 86.2±0.4 | 80.8±0.6 | 89.4±0.5 | 89.0±0.4 | 88.9±0.4 | 95.1±0.2 | 92.3±0.3 | 89.7±0.3 | 88.4 |
| Rot+Trans [25] | ResNet-18 | 80.4±0.3 | 96.4±0.2 | 85.9±0.3 | 81.1±0.5 | 91.3±0.3 | 89.6±0.3 | 89.9±0.3 | 95.9±0.1 | 95.0±0.1 | 92.6±0.2 | 89.8 |
| GOAD [2] | ResNet-18 | 75.5±0.3 | 94.1±0.3 | 81.8±0.5 | 72.0±0.3 | 83.7±0.9 | 84.4±0.3 | 82.9±0.8 | 93.9±0.3 | 92.9±0.3 | 89.5±0.2 | 85.1 |
| CSI (ours) | ResNet-18 | **89.9**±0.1 | **99.1**±0.0 | **93.1**±0.2 | **86.4**±0.2 | **93.9**±0.1 | **93.2**±0.2 | **95.1**±0.1 | **98.7**±0.0 | **97.9**±0.0 | **95.5**±0.1 | **94.3** |

(b) One-class CIFAR-100 (super-class)

| Method | Network | AUROC |
|---|---|---|
| OC-SVM$^*$ [59] | - | 63.1 |
| Geom$^*$ [15] | WRN-16-8 | 78.7 |
| Rot [25] | ResNet-18 | 77.7 |
| Rot+Trans [25] | ResNet-18 | 79.8 |
| GOAD [2] | ResNet-18 | 74.5 |
| CSI (ours) | ResNet-18 | **89.6** |

(c) One-class ImageNet-30

| Method | Network | AUROC |
|---|---|---|
| Rot$^*$ [25] | ResNet-18 | 65.3 |
| Rot+Trans$^*$ [25] | ResNet-18 | 77.9 |
| Rot+Attn$^*$ [25] | ResNet-18 | 81.6 |
| Rot+Trans+Attn$^*$ [25] | ResNet-18 | 84.8 |
| Rot+Trans+Attn+Resize$^*$ [25] | ResNet-18 | 85.7 |
| CSI (ours) | ResNet-18 | **91.6** |

After training the embedding network $f_\theta(x)$ with the SupCLR objective (10), we train a linear classifier upon $f_\theta(x)$ to model $p_{\text{SupCLR}}(y|x)$.

**Supervised extension of CSI.** We extend CSI by incorporating the shifting transformations $\mathcal{S}$ into the SupCLR objective: here, we consider a joint label $(y, y^{\mathcal{S}}) \in \mathcal{Y} \times \mathcal{S}$ of class label $y$ and shifting transformation $y^{\mathcal{S}}$. Then, the *supervised contrasting shifted instances* (sup-CSI) loss is given by:

$$\mathcal{L}_{\text{sup-CSI}} := \mathcal{L}_{\text{SupCLR}} \left( \bigcup_{S \in \mathcal{S}} \mathcal{C}_S; \mathcal{T} \right), \quad \text{where } \mathcal{C}_S := \{(S(x_i), (y_i, S))\}_{i=1}^B. \tag{11}$$

Note that we do not use the auxiliary classification loss $\mathcal{L}_{\text{cls-SI}}$ (4), since the objective already classifies the shifted instances under a *self-label augmented* [35] space $\mathcal{Y} \times \mathcal{S}$.

Upon the learned representation via (11), we additionally train two linear classifiers: $p_{\text{CSI}}(y|x)$ and $p_{\text{CSI-joint}}(y, y^{\mathcal{S}}|x)$ that predicts the class labels and joint labels, respectively. We directly apply $s_{\text{sup}}(x)$ for the former $p_{\text{CSI}}(y|x)$. For the latter, on the other hand, we marginalize the joint prediction over the shifting transformation in a similar manner of Section 2.3. Precisely, let $l(x) \in \mathbb{R}^{C \times K}$ be logit values of $p_{\text{CSI-joint}}(y, y^{\mathcal{S}}|x)$ for $|\mathcal{Y}| = C$ and $|\mathcal{S}| = K$, and $l(x)_k \in \mathbb{R}^C$ be logit values correspond to $p_{\text{CSI-joint}}(y, y^{\mathcal{S}} = S_k|x)$. Then, the ensembled probability is:

$$p_{\text{CSI-ens}}(y|x) := \sigma \left( \frac{1}{K} \sum_k l(S_k(x))_k \right), \tag{12}$$

where $\sigma$ denotes the softmax activation. Here, we use $p_{\text{CSI-ens}}$ to compute the confidence $s_{\text{sup}}(x)$. We denote the confidence computed by $p_{\text{CSI}}$ and $p_{\text{CSI-ens}}$ and "CSI" and "CSI-ens", respectively.

## 3 Experiments

In Section 3.1, we report OOD detection results on unlabeled one-class, unlabeled multi-class, and labeled multi-class datasets. In Section 3.2, we analyze the effects on various shifting transformations in the context of OOD detection, as well as an ablation study on each component we propose.

Table 2: AUROC (%) of various OOD detection methods trained on unlabeled (a) CIFAR-10 and (b) ImageNet-30. The reported results are averaged over five trials, subscripts denote standard deviation, and bold denote the best results. * denotes the values from the reference.

(a) Unlabeled CIFAR-10

| Method | Network | CIFAR10 → | | | | | | |
| --- | --- | --- | --- | --- | --- | --- | --- | --- |
| | | SVHN | LSUN | ImageNet | LSUN (FIX) | ImageNet (FIX) | CIFAR-100 | Interp. |
| Likelihood* | PixelCNN++ | 8.3 | - | 64.2 | - | - | 52.6 | 52.6 |
| Likelihood* | Glow | 8.3 | - | 66.3 | - | - | 58.2 | 58.2 |
| Likelihood* | EBM | 63.0 | - | - | - | - | - | 70.0 |
| Likelihood Ratio* [55] | PixelCNN++ | 91.2 | - | - | - | - | - | - |
| Input Complexity* [61] | PixelCNN++ | 92.9 | - | 58.9 | - | - | 53.5 | - |
| Input Complexity* [61] | Glow | 95.0 | - | 71.6 | - | - | 73.6 | - |
| Rot [25] | ResNet-18 | $97.6_{\pm0.2}$ | $89.2_{\pm0.7}$ | $90.5_{\pm0.3}$ | $77.7_{\pm0.3}$ | $83.2_{\pm0.1}$ | $79.0_{\pm0.1}$ | $64.0_{\pm0.3}$ |
| Rot+Trans [25] | ResNet-18 | $97.8_{\pm0.2}$ | $92.8_{\pm0.9}$ | $94.2_{\pm0.7}$ | $81.6_{\pm0.4}$ | $86.7_{\pm0.1}$ | $82.3_{\pm0.2}$ | $68.1_{\pm0.8}$ |
| GOAD [2] | ResNet-18 | $96.3_{\pm0.2}$ | $89.3_{\pm1.5}$ | $91.8_{\pm1.2}$ | $78.8_{\pm0.3}$ | $83.3_{\pm0.1}$ | $77.2_{\pm0.3}$ | $59.4_{\pm1.1}$ |
| CSI (ours) | ResNet-18 | $\mathbf{99.8}_{\pm0.0}$ | $\mathbf{97.5}_{\pm0.3}$ | $\mathbf{97.6}_{\pm0.3}$ | $\mathbf{90.3}_{\pm0.3}$ | $\mathbf{93.3}_{\pm0.1}$ | $\mathbf{89.2}_{\pm0.1}$ | $\mathbf{79.3}_{\pm0.2}$ |

(b) Unlabeled ImageNet-30

| Method | Network | ImageNet-30 → | | | | | | | |
| --- | --- | --- | --- | --- | --- | --- | --- | --- | --- |
| | | CUB-200 | Dogs | Pets | Flowers | Food-101 | Places-365 | Caltech-256 | DTD |
| Rot [25] | ResNet-18 | $76.5_{\pm0.7}$ | $77.2_{\pm0.5}$ | $70.0_{\pm0.5}$ | $87.2_{\pm0.2}$ | $72.7_{\pm1.5}$ | $52.6_{\pm1.4}$ | $70.9_{\pm0.1}$ | $89.9_{\pm0.5}$ |
| Rot+Trans [25] | ResNet-18 | $74.5_{\pm0.5}$ | $77.8_{\pm1.1}$ | $70.0_{\pm0.8}$ | $86.3_{\pm0.3}$ | $71.6_{\pm1.4}$ | $53.1_{\pm1.7}$ | $70.0_{\pm0.2}$ | $89.4_{\pm0.6}$ |
| GOAD [2] | ResNet-18 | $71.5_{\pm1.4}$ | $74.3_{\pm1.6}$ | $65.5_{\pm1.3}$ | $82.8_{\pm1.4}$ | $68.7_{\pm0.7}$ | $51.0_{\pm1.1}$ | $67.4_{\pm0.8}$ | $87.5_{\pm0.8}$ |
| CSI (ours) | ResNet-18 | $\mathbf{90.5}_{\pm0.1}$ | $\mathbf{97.1}_{\pm0.1}$ | $\mathbf{85.2}_{\pm0.2}$ | $\mathbf{94.7}_{\pm0.4}$ | $\mathbf{89.2}_{\pm0.3}$ | $\mathbf{78.3}_{\pm0.3}$ | $\mathbf{87.1}_{\pm0.1}$ | $\mathbf{96.9}_{\pm0.1}$ |

**Setup.** We use ResNet-18 [20] architecture for all the experiments. For data augmentations $\mathcal{T}$, we adopt those used by Chen et al. [5]: namely, we use the combination of Inception crop [64], horizontal flip, color jitter, and grayscale. For shifting transformations $\mathcal{S}$, we use the random rotation $0°, 90°, 180°, 270°$ unless specified otherwise, as rotation has the highest OOD-ness (see Section 2.2) values for natural images, *e.g.*, CIFAR-10 [33]. However, we remark that the best shifting transformation can be different for other datasets, *e.g.*, Gaussian noise performs better than rotation for texture datasets (see Table 6 in Section 3.2). By default, we train our models from scratch with the training objective in (5) and detect OOD samples with the ensembled version of the score in (9).

We mainly report the area under the receiver operating characteristic curve (AUROC) as a threshold-free evaluation metric for a detection score. In addition, we report the test accuracy and the expected calibration error (ECE) [45, 19] for the experiments on labeled multi-class datasets. Here, ECE estimates whether a classifier can indicate when they are likely to be incorrect for test samples (from in-distribution) by measuring the difference between prediction confidence and accuracy. The formal description of the metrics and detailed experimental setups are in Appendix A.

### 3.1 Main results

**Unlabeled one-class datasets.** We start by considering the *one-class* setup: here, for a given multi-class dataset of $C$ classes, we conduct $C$ one-class classification tasks, where each task chooses one of the classes as in-distribution while the remaining classes being out-of-distribution. We run our experiments on three datasets, following the prior work [15, 25, 2]: CIFAR-10 [33], CIFAR-100 labeled into 20 super-classes [33], and ImageNet-30 [25] datasets. We compare CSI with various prior methods including one-class classifier [59, 56], reconstruction-based [58, 52], and self-supervised [15, 25, 2] approaches. Table 1 summarizes the results, showing that CSI significantly outperforms the prior methods in all the tested cases. We provide the full, additional results, *e.g.*, class-wise AUROC on CIFAR-100 (super-class) and ImageNet-30, in Appendix C.

**Unlabeled multi-class datasets.** In this setup, we assume that in-distribution samples are from a specific multi-class dataset without labels, testing on various external datasets as out-of-distribution. We compare CSI on two in-distribution datasets: CIFAR-10 [33] and ImageNet-30 [25]. We consider the following datasets as out-of-distribution: SVHN [48], resized LSUN and ImageNet [39], CIFAR-100 [33], and linearly-interpolated samples of CIFAR-10 (Interp.) [11] for CIFAR-10 experiments, and CUB-200 [67], Dogs [29], Pets [51], Flowers [49], Food-101 [3], Places-365 [75], Caltech-256 [18], and DTD [8] for ImageNet-30. We compare CSI with various prior methods, including density-based [11, 55, 61] and self-supervised [15, 2] approaches.

Table 3: Test accuracy (%), ECE (%), and AUROC (%) of confidence-calibrated classifiers trained on labeled (a) CIFAR-10 and (b) ImageNet-30. The reported results are averaged over five trials for CIFAR-10 and one trial for ImageNet-30. Subscripts denote standard deviation, and bold denote the best results. CSI-ens denotes the ensembled prediction, *i.e.*, 4 times slower (as we use rotation).

(a) Labeled CIFAR-10

| Train method | Test acc. | ECE | CIFAR10 → | | | | | | |
|---|---|---|---|---|---|---|---|---|---|
| | | | SVHN | LSUN | ImageNet | LSUN (FIX) | ImageNet (FIX) | CIFAR100 | Interp. |
| Cross Entropy | $93.0_{\pm0.2}$ | $6.44_{\pm0.2}$ | $88.6_{\pm0.9}$ | $90.7_{\pm0.5}$ | $88.3_{\pm0.6}$ | $87.5_{\pm0.3}$ | $87.4_{\pm0.3}$ | $85.8_{\pm0.3}$ | $75.4_{\pm0.7}$ |
| SupCLR [30] | $93.8_{\pm0.1}$ | $5.56_{\pm0.1}$ | $97.3_{\pm0.1}$ | $92.8_{\pm0.5}$ | $91.4_{\pm1.2}$ | $91.6_{\pm1.5}$ | $90.5_{\pm0.5}$ | $88.6_{\pm0.2}$ | $75.7_{\pm0.1}$ |
| CSI (ours) | $94.8_{\pm0.1}$ | $4.40_{\pm0.1}$ | $96.5_{\pm0.2}$ | $96.3_{\pm0.5}$ | $96.2_{\pm0.4}$ | $92.1_{\pm0.5}$ | $92.4_{\pm0.0}$ | $90.5_{\pm0.1}$ | $78.5_{\pm0.2}$ |
| CSI-ens (ours) | $\mathbf{96.1}_{\pm0.1}$ | $\mathbf{3.50}_{\pm0.1}$ | $\mathbf{97.9}_{\pm0.1}$ | $\mathbf{97.7}_{\pm0.4}$ | $\mathbf{97.6}_{\pm0.3}$ | $\mathbf{93.5}_{\pm0.4}$ | $\mathbf{94.0}_{\pm0.1}$ | $\mathbf{92.2}_{\pm0.1}$ | $\mathbf{80.1}_{\pm0.3}$ |

(b) Labeled ImageNet-30

| Train method | Test acc. | ECE | ImageNet-30 → | | | | | | | |
|---|---|---|---|---|---|---|---|---|---|---|
| | | | CUB-200 | Dogs | Pets | Flowers | Food-101 | Places-365 | Caltech-256 | DTD |
| Cross Entropy | 94.3 | 5.08 | 88.0 | 96.7 | 95.0 | 89.7 | 79.8 | 90.5 | 90.6 | 90.1 |
| SupCLR [30] | 96.9 | 3.12 | 86.3 | 95.6 | 94.2 | 92.2 | 81.2 | 89.7 | 90.2 | 92.1 |
| CSI (ours) | 97.0 | 2.61 | 93.4 | 97.7 | 96.9 | 96.0 | 87.0 | 92.5 | 91.9 | 93.7 |
| CSI-ens (ours) | **97.8** | **2.19** | **94.6** | **98.3** | **97.4** | **96.2** | **88.9** | **94.0** | **93.2** | **97.4** |

Table 2 shows the results. Overall, CSI significantly outperforms the prior methods in all benchmarks tested. We remark that CSI is particularly effective for detecting hard (*i.e.*, near-distribution) OOD samples, *e.g.*, CIFAR-100 and Interp. in Table 2a. Also, CSI still shows a notable performance in the cases when prior methods often fail, *e.g.*, AUROC of 50% (*i.e.*, random guess) for Places-365 dataset in Table 2b. Finally, we notice that the resized LSUN and ImageNet datasets officially released by Liang et al. [39] might be misleading to evaluate detection performance for hard OODs: we find that those datasets contain some unintended artifacts, due to incorrect resizing procedure. Such an artifact makes those datasets easily-detectable, *e.g.*, via input statistics. In this respect, we produce and test on their fixed versions, coined LSUN (FIX), and ImageNet (FIX). See Appendix I for details.

**Labeled multi-class datasets.** We also consider the *labeled* version of the above setting: namely, we now assume that every in-distribution sample also contains discriminative label information. We use the same datasets considered in the unlabeled multi-class setup for in- and out-of-distribution datasets. We train our model as proposed in Section 2.4, and compare it with those trained by other methods, the cross-entropy and supervised contrastive learning (SupCLR) [30]. Since our goal is to calibrate the confidence, the maximum softmax probability is used to detect OOD samples (see [22]).

Table 3 shows the results. Overall, CSI consistently improves AUROC and ECE for all benchmarks tested. Interestingly, CSI also improves test accuracy; even our original purpose of CSI is to learn a representation for OOD detection. CSI can further improve the performance by ensembling over the transformations. We also remark that our results on unlabeled datasets (in Table 2) already show comparable performance to the supervised baselines (in Table 3).

## 3.2 Ablation study

We perform an ablation study on various shifting transformations, training objectives, and detection scores. Throughout this section, we report the mean AUROC values on one-class CIFAR-10.

**Shifting transformation.** We measure the OOD-ness (see Section 2.2) of transformations, *i.e.*, the AUROC between in-distribution vs. transformed samples under vanilla SimCLR, and the effects of those transformations when used as a shifting transformation. In particular, we consider Cutout [10], Sobel filtering [28], Gaussian noise, Gaussian blur, and rotation [14]. We remark that these transformations are reported to be ineffective in improving the class discriminative power of SimCLR [5]. We also consider the transformation coined "Perm", which randomly permutes each part of the evenly partitioned image. Intuitively, such transformations commonly *shift* the input distribution, hence forcing them to be *aligned* can be harmful. Figure 1 visualizes the considered transformations.

Table 4 shows AUROC values of the vanilla SimCLR, where the in-distribution samples shifted by the chosen transformation are given as OOD samples. The shifted samples are easily detected: it validates our intuition that the considered transformations *shift* the input distribution. In particular, "Perm" and "Rotate" are the most distinguishable, which implies they shift the distribution the most.

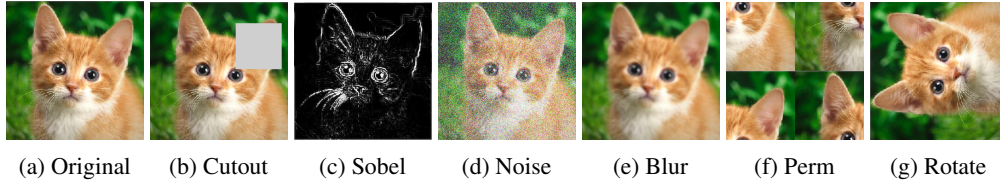

|  (a) Original | (b) Cutout | (c) Sobel | (d) Noise | (e) Blur | (f) Perm | (g) Rotate |

Figure 1: Visualization of the original image and the considered shifting transformations.

Table 4: OOD-ness (%), *i.e.*, the AUROC between in-distribution vs. transformed samples under the vanilla SimCLR (see Section 2.2), of various transformations. The vanilla SimCLR is trained on one-class CIFAR-10 under ResNet-18. Each column denotes the applied transformation.

|  | Cutout | Sobel | Noise | Blur | Perm | Rotate |
|---|---|---|---|---|---|---|
| OOD-ness | 79.5 | 69.2 | 74.4 | 76.0 | 83.8 | 85.2 |

Table 5: Ablation study on various transformations, added or removed from the vanilla SimCLR. "Align" and "Shift" indicates that the transformation is used as $\mathcal{T}$ and $\mathcal{S}$, respectively. (a) We add a new transformation as an aligned (up) or shifting (down) transformations. (b) We remove (up) or convert-to-shift (down) the transformation from the vanilla SimCLR. All reported values are the mean AUROC (%) over one-class CIFAR-10, and "Base" denotes the vanilla SimCLR.

| (a) Add transformations | | | | | | | | (b) Remove transformations | | | |
|---|---|---|---|---|---|---|---|---|---|---|---|
| Base | | Cutout | Sobel | Noise | Blur | Perm | Rotate | | Crop | Jitter | Gray |
| 87.9 | +Align | 84.3 | 85.0 | 85.5 | 88.0 | 73.1 | 76.5 | -Align | 55.7 | 78.8 | 78.4 |
|  | +Shift | 88.5 | 88.3 | 89.3 | 89.2 | 90.7 | 94.3 | +Shift | - | - | 88.3 |

Note that "Perm" and "Rotate" turns out to be the most effective shifting transformations; it implies that the transformations *shift* the distribution most indeed performs best for CSI.[3]

Besides, we apply the transformation upon the vanilla SimCLR: align the transformed samples to the original samples (*i.e.*, use as $\mathcal{T}$) or consider them as the shifted samples (*i.e.*, use as $\mathcal{S}$). Table 5a shows that aligning the transformations degrade (or on par) the detection performance, while shifting the transformations gives consistent improvements. We also remove or convert-to-shift the transformation from the vanilla SimCLR in Table 5b, and see similar results. We remark that one can further improve the performance by combining multiple shifting transformations (see Appendix G).

**Data-dependence of shifting transformations.** We remark that the best shifting transformation depends on the dataset. For example, consider the rotation-invariant datasets: Describable Textures Dataset (DTD) [8] and Textile [60] are in- vs. out-of-distribution, respectively (see Appendix J for more visual examples). For such datasets, rotation (Rot.) does not shift the distribution, and Gaussian noise (Noise) is more suitable transformation (see Table 6a). Table 6b shows that CSI using Gaussian noise ("CSI(N)") indeed improves the vanilla SimCLR ("Base") while CSI using rotation ("CSI(R)") degrades instead. This results support our principles on selecting shifting transformations.

Table 6: OOD-ness (%) and AUROC (%) on DTD, where Textile is used for OOD.

| (a) OOD-ness | | (b) AUROC | | |
|---|---|---|---|---|
| Rot. | Noise | Base | CSI(R) | CSI(N) |
| 50.6 | **75.7** | 70.3 | 65.9 | **80.1** |

**Linear evaluation.** We also measure the linear evaluation [32], the accuracy of a linear classifier to discriminate classes of in-distribution samples. It is widely used for evaluating the quality of (unsupervised) learned representation. We report the linear evaluation of vanilla SimCLR and CSI (with shifting rotation), trained under unlabeled CIFAR-10. They show comparable results, 90.48% for SimCLR and 90.19% for CSI; CSI is more specialized to learn a representation for OOD detection.

**Training objective.** In Table 7a, we assess the individual effects of each component that consists of our final training objective (5): namely, we compare the vanilla SimCLR (2), contrasting shifted

Table 7: Ablation study on each component of our proposed (a) training objective and (b) detection score. For (a), we use the corresponding detection score for each training loss; namely, (6) to (9) for (2) to (5), respectively. For (b), we use the model trained by the final training loss (5). We measure the mean AUROC (%) values, trained under CIFAR-10 with ResNet-18. Each row indicates the corresponding equation of the given checkmarks, and bold denotes the best results. "Con.", "Cls.", and "Ensem." denotes contrast, classify, and ensemble, respectively.

| (a) Training objective | | | | | | (b) Detection score | | | | |
|---|---|---|---|---|---|---|---|---|---|---|
| | SimCLR | Con. | Cls. | AUROC | | | Con. | Cls. | Ensem. | AUROC |
| $\mathcal{L}_{\text{SimCLR}}$ (2) | ✓ | - | - | 87.9 | | $s_{\text{con}}$ (6) | ✓ | - | - | 91.3 |
| $\mathcal{L}_{\text{con-SI}}$ (3) | ✓ | ✓ | - | 91.6 | | $s_{\text{con-SI}}$ (7) | ✓ | - | ✓ | 93.3 |
| $\mathcal{L}_{\text{cls-SI}}$ (4) | - | - | ✓ | 88.6 | | $s_{\text{cls-SI}}$ (8) | - | ✓ | ✓ | 93.8 |
| $\mathcal{L}_{\text{CSI}}$ (5) | ✓ | ✓ | ✓ | **94.3** | | $s_{\text{CSI}}$ (9) | ✓ | ✓ | ✓ | **94.3** |

instances (3), and classifying shifted instances (4) losses. For the evaluation of the models of different training objectives (2) to (5), we use the detection scores defined in (6) to (9), respectively. We remark that both contrasting and classifying shows better results than the vanilla SimCLR; and combining them (*i.e.*, the final CSI objective (5)) gives further improvements, *i.e.*, two losses are complementary.

**Detection score.** Finally, Table 7b shows the effect of each component in our detection score: the vanilla contrastive (6), contrasting shifted instances (7), and classifying shifted instances (8) scores. We ensemble the scores over both $\mathcal{T}$ and $\mathcal{S}$ for (7) to (9), and use a single sample for (6). All the reported values are evaluated from the model trained by the final objective 5. Similar to above, both contrasting and classifying scores show better results than the vanilla contrastive score; and combining them (*i.e.*, the final CSI score (9)) gives further improvements.

## 4 Related work

**OOD detection.** Recent works on unsupervised OOD detection (*i.e.*, no external OOD samples) [26] can be categorized as: (a) density-based [74, 46, 6, 47, 11, 55, 61, 17], (b) reconstruction-based [58, 76, 9, 54, 52, 7], (c) one-class classifier [59, 56], and (d) self-supervised [15, 25, 2] approaches. Our work falls into (c) the self-supervised approach, as it utilizes the representation learned from self-supervision [14]. However, unlike prior works [15, 25, 2] focusing on the self-label classification tasks (*e.g.*, predict the angle of the rotated image), we first incorporate *contrastive learning* [5] for OOD detection. Concurrently, Winkens et al. [68] and Liu and Abbeel [40] report that contrastive learning also improves the OOD detection performance of classifiers [39, 38, 25].

**Confidence-calibrated classifiers.** Confidence-calibrated classifiers aim to calibrate the prediction confidence (maximum softmax probability), which can be directly used as an uncertainty estimator for both within in-distribution [45, 19] and in- vs. out-of-distribution [22, 37]. Prior works improved calibration through inference [19] or training [37] schemes, which are can be jointed applied to our method. Some works design a specific detection score upon the pre-trained classifiers [39, 38], but they only target OOD detection, while ours also consider the in-distribution calibration.

**Self-supervised learning.** Self-supervised learning [14, 32], particularly contrastive learning [13] via instance discrimination [69], has shown remarkable success on visual representation learning [21, 5]. However, most prior works focus on the downstream tasks (*e.g.*, classification), and other advantages (*e.g.*, uncertainty or robustness) are rarely investigated [25, 31]. Our work, concurrent with [40, 68], first verifies that contrastive learning is also effective for OOD detection. In particular, we find that the shifting transformations, which were known to be harmful and unused for the standard contrastive learning [5], can help OOD detection. This observation provides new considerations for selecting transformations, *i.e.*, which transformation should be used for positive or negative [66, 71].

We further provide a more comprehensive survey and discussions with prior works in Appendix B.

## 5 Conclusion

We propose a simple yet effective method named contrasting shifted instances (CSI), which extends the power of contrastive learning for out-of-distribution (OOD) detection problems. CSI demonstrates outstanding performance under various OOD detection scenarios. We believe our work would guide various future directions in OOD detection and self-supervised learning as an important baseline.

## Acknowledgements

This work was supported by Institute of Information & Communications Technology Planning & Evaluation (IITP) grant funded by the Korea government (MSIT) (No.2019-0-00075, Artificial Intelligence Graduate School Program (KAIST) and No.2017-0-01779, A machine learning and statistical inference framework for explainable artificial intelligence). We thank Sihyun Yu, Chaewon Kim, Hyuntak Cha, Hyunwoo Kang, and Seunghyun Lee for helpful feedback and suggestions.

## Broader Impact

This paper is focused on the subject of *out-of-distribution (OOD)* (or novelty, anomaly) detection, which is an essential ingredient for building safe and reliable intelligent systems [1]. We expect our results to have two consequences for academia and broader society.

**Rethinking representation for OOD detection.** In this paper, we demonstrate that the representation for classification (or other related tasks, measured by linear evaluation [32]) can be different from the representation for OOD detection. In particular, we verify that the "hard" augmentations, thought to be harmful for contrastive representation learning [5], can be helpful for OOD detection. Our observation raises new questions for both representation learning and OOD detection: (a) representation learning researches should also report the OOD detection results as an evaluation metric, (b) OOD detection researches should more investigate the specialized representation.

**Towards reliable intelligent system.** The intelligent system should be robust to the potential dangers of uncertain environments (*e.g.*, financial crisis [65]) or malicious adversaries (*e.g.*, cybersecurity [34]). Detecting outliers is also related to human safety (*e.g.*, medical diagnosis [4] or autonomous driving [12]), and has a broad range of industrial applications (*e.g.*, manufacturing inspection [42]). However, the system can be stuck into *confirmation bias*, *i.e.*, ignore new information with a myopic perspective. We hope the system to balance the exploration and exploitation of the knowledge.

## Footnotes

[2]We do not compare with methods using *external* OOD samples [24, 57].

[3]We also have tried contrasting *external* OOD samples similarly to [24]; however, we find that naïvely using them in our framework degrade the performance. This is because the contrastive loss also discriminates *within* external OOD samples, which is unnecessary and an additional learning burden for our purpose.

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
