[Supplementary Material]

# Appendix

## CSI: Novelty Detection via Contrastive Learning on Distributionally Shifted Instances

## A  Experimental details

**Training details.** We use ResNet-18 [20] as the base encoder network $f_\theta$ and 2-layer multi-layer perceptron with 128 embedding dimension as the projection head $g_\phi$. All models are trained by minimizing the final loss $\mathcal{L}_{\texttt{CSI}}$ (5) with a temperature of $\tau = 0.5$. We follow the same optimization step of SimCLR [5]. For optimization, we train CSI with 1,000 epoch under LARS optimizer [72] with weight decay of $1e-6$ and momentum with 0.9. For the learning rate scheduling, we use linear warmup [16] for early 10 epochs until learning rate of 1.0 and decay with cosine decay schedule without a restart [41]. We use batch size of 512 for both vanilla SimCLR and ours: where the batch is given by $\mathcal{B}$ for vanilla SimCLR and the aggregated one $\bigcup_{S \in \mathcal{S}} \mathcal{B}_S$ for ours. Furthermore, we use global batch normalization (BN) [27], which shares the BN parameters (mean and variance) over the GPUs in distributed training.

For supervised contrastive learning (SupCLR) [30] and supervised CSI, we select the best temperature from $\{0.07, 0.5\}$: SupCLR recommend 0.07 but 0.5 was better in our experiments. For training the encoder $f_\theta$, we use the same optimization scheme as above, except using 700 for the epoch. For training the linear classifier, we train the model for 100 epochs with batch size 128, using stochastic gradient descent with momentum 0.9. The learning rate starts at 0.1 and is dropped by a factor of 10 at 60%, 75%, and 90% of the training progress.

**Data augmentation details.** We use SimCLR augmentations: Inception crop [64], horizontal flip, color jitter, and grayscale for random augmentations $\mathcal{T}$, and rotation as shifting transformation $\mathcal{S}$. The detailed description of the augmentations are as follows:

- **Inception crop.** Randomly crops the area of the original image with uniform distribution 0.08 to 1.0. After the crop, cropped image are resized to the original image size.
- **Horizontal flip.** Flips the image horizontally with 50% of probability.
- **Color jitter.** Change the hue, brightness, and saturation of the image. We transform the RGB (red, green, blue) image into an HSV (hue, saturation, value) image format and add noise to the HSV channels. We apply color jitter with 80% of probability.
- **Grayscale.** Convert into a gray image. Randomly apply a grayscale with 20% of probability.
- **Rotation.** We use rotation as $\mathcal{S}$, the shifting transformation, $\{0°, 90°, 180°, 270°\}$. For a given batch $\mathcal{B}$, we apply each rotation degree to obtain the new batch for CSI: $\bigcup_{S \in \mathcal{S}} \mathcal{B}_S$.

| (a) Original | (b) Inception crop | (c) Horizontal flip | (d) Color jitter | (e) Grayscale |

Figure 2: Visualization of original image and SimCLR augmentations.

**Dataset details.** For one-class datasets, we train one class of CIFAR-10 [33], CIFAR-100 (super-class) [33], and ImageNet-30 [25]. CIFAR-10 and CIFAR-100 consist of 50,000 training and 10,000 test images with 10 and 20 (super-class) image classes, respectively. ImageNet-30 contains 39,000 training and 3,000 test images with 30 image classes.

For unlabeled and labeled multi-class datasets, we train ResNet with CIFAR-10 and ImageNet-30. For CIFAR-10, out-of-distribution (OOD) samples are as follows: SVHN [48] consists of 26,032 test images with 10 digits, resized LSUN [39] consists of 10,000 test images of 10 different scenes, resized ImageNet [39] consists of 10,000 test images with 200 images classes from a subset of full ImageNet dataset, Interp. consists of 10,000 test images of linear interpolation of CIFAR-10 test images, and LSUN (FIX), ImageNet (FIX) consists of 10,000 test images, respectively with following details in Appendix I. For multi-class ImageNet-30, OOD samples are as follows: CUB-200 [67], Stanford Dogs [29], Oxford Pets [51], Oxford Flowers [49], Food-101 [3] without the "hotdog" class to avoid overlap, Places-365 [75] with small images (256 * 256) validation set, Caltech-256 [18], and Describable Textures Dataset (DTD) [8]. Here, we randomly sample 3,000 images to balance with the in-distribution test set.

**Evaluation metrics.** For evaluation, we measure the two metrics that each measures (a) the effectiveness of the proposed score in distinguishing in- and out-of-distribution images, (b) the confidence calibration of softmax classifier.

- **Area under the receiver operating characteristic curve (AUROC).** Let TP, TN, FP, and FN denote true positive, true negative, false positive and false negative, respectively. The ROC curve is a graph plotting true positive rate = TP / (TP+FN) against the false positive rate = FP / (FP+TN) by varying a threshold.

- **Expected calibration error (ECE).** For a given test data $\{(x_n, y_n)\}_{n=1}^N$, we group the predictions into $M$ interval bins (each of size $1/M$). Let $B_m$ be the set of indices of samples whose prediction confidence falls into the interval $(\frac{m-1}{M}, \frac{m}{M}]$. Then, the expected calibration error (ECE) [45, 19] is follows:

$$\text{ECE} = \sum_{m=1}^{M} \frac{|B_m|}{N} |\text{acc}(B_m) - \text{conf}(B_m)|, \tag{13}$$

where $\text{acc}(B_m)$ is accuracy of $B_m$: $\text{acc}(B_m) = \frac{1}{|B_m|} \sum_{i \in B_m} \mathbb{1}_{\{y_i = \arg\max_y p(y|x_i)\}}$ where $\mathbb{1}$ is indicator function and $\text{conf}(B_m)$ is confidence of $B_m$: $\text{conf}(B_m) = \frac{1}{|B_m|} \sum_{i \in B_m} q(x_i)$ where $q(x_i)$ is the confidence of data $x_i$.

# B    Detailed review on related work

## B.1    OOD detection

Out-of-distribution (OOD) detection is a classic and essential problem in machine learning, studied under different names, *e.g.*, novelty or anomaly detection [26]. In this paper, we primarily focus on *unsupervised* OOD detection, which is arguably the most traditional and popular setup in the field [59]. In this setting, the detector can only access in-distribution samples while required to identify unseen OOD samples. There are other settings, *e.g.*, semi-supervised setting - the detector can access a small subset of out-of-distribution samples [24, 57], or supervised setting - the detector knows the target out-of-distribution, but we do not consider those settings in this paper. We remark that the unsupervised setting is the most practical and challenging scenario since there are *infinitely* many cases for out-of-distribution, and it is often not possible to have such external data.

Most recent works can be categorized as: (a) density-based [74, 46, 6, 47, 11, 55, 61, 17], (b) reconstruction-based [58, 76, 9, 54, 52, 7], (c) one-class classifier [59, 56, 57], and (d) self-supervised [15, 25, 2] approaches. We note that there are more extensive literature on this topic, but we mainly focus on the recent work based on deep learning. Brief description for each method are as follows:

- **Density-based methods.** Density-based methods are one of the most classic and principled approaches for OOD detection. Intuitively, they directly use the likelihood of the sample as the detection score. However, recent studies reveal that the likelihood is often not the best metric - especially for deep neural networks with complex datasets [46]. Several work thus proposed modified scores, *e.g.*, typicality [47], WAIC [6], likelihood ratio [55], input complexity [61], or unnormalized likelihood (*i.e.*, energy) [11, 17].

- **Reconstruction-based methods.** Reconstruction-based approach is another popular line of research for OOD detection. It trains an encoder-decoder network that reconstructs the training data in an unsupervised manner. Since the network would less generalize for unseen OOD samples, they use the reconstruction loss as a detection score. Some works utilize auto-encoders [76, 54] or generative adversarial networks [58, 9, 52].

- **One-class classifiers.** One-class classifiers are also a classic and principled approach for OOD detection. They learn a decision boundary of in- vs. out-of-distribution samples by giving some margin covering the in-distribution samples [59]. Recent works have shown that the one-class classifier is effective upon the deep representation [56].

- **Self-supervised methods.** Self-supervised approaches are a relatively new technique based on the rich representation learned from self-supervision [14]. They train a network with a pre-defined task (*e.g.*, predict the angle of the rotated image) on the training set, and use the generalization error to detect OOD samples. Recent self-supervised approaches show outstanding results on various OOD detection benchmark datasets [15, 25, 2].

Our work falls into (c) the self-supervised approach [15, 25, 2]. However, unlike prior work focusing on the self-label classification tasks (*e.g.*, rotation [14]) which trains an auxiliary classifier to predict the transformation applied to the sample, we first incorporate *contrastive learning* [5] for OOD detection. To that end, we design a novel detection score utilizing the unique characteristic of contrastive learning, *e.g.*, the features in the projection layer learned by cosine similarity. We also propose a novel self-supervised training scheme that further improves the representation for OOD detection. Nevertheless, we acknowledge that the prior work largely inspired our work. For instance, the classifying shifted instances loss (4) follows the form of auxiliary classifiers [25], which gives further improvement upon our novel contrasting shifted instances loss (3).

Concurrently, Winkens et al. [68] and Liu and Abbeel [40] report the similar observations that contrastive learning also improves the OOD detection performance of classifiers [39, 38, 25]. Winkens et al. [68] jointly train a classifier with the SimCLR [5] objective and use the Mahalanobis distance [38] as a detection score. Liu and Abbeel [40] approximates JEM [17] (a joint model of classifier and energy-based model [11]) by a combination of classification and contrastive loss and use density-based detection scores [17]. In contrast to both work, we mainly focus on the *unlabeled OOD* setting (although we also discuss the confident-calibrated classifiers). Here, we design a novel detection score, since how to utilize the contrastive representation (which is learned in an unsupervised manner) for OOD detection have not been explored before.

## B.2 Confidence-calibrated classifiers

Another line of research is on confidence-calibrated classifiers [22], which relaxes the overconfidence issue of the classifiers. There are two types of calibration: (a) *in-distribution* calibration [45, 19], that aligns the uncertainty and the actual accuracy, measured by ECE, and (b) *out-of-distribution* detection [22, 37], that reduces the uncertainty of OOD samples, measured by AUROC. Note that the goal of confidence-calibrated classifiers is to regularize the prediction. Hence, the softmax probability is used for all three tasks: classification, in-distribution calibration, and out-of-distribution detection. Namely, the detection score is given by the prediction confidence (or maximum softmax probability) [22]. Prior works improved calibration through inference (temperature scaling) [19] or training (regularize predictions of OOD samples) [37] schemes, which can be jointly applied to our method. Some works design a specific detection score upon the pre-trained classifiers [39, 38], but they only target OOD detection, while ours also consider the in-distribution calibration.

## B.3 Self-supervised learning

Self-supervised learning [14, 32] has shown remarkable success in learning representations. In particular, contrastive learning [13] via instance discrimination [69] show the state-of-the-art results on visual representation learning [21, 5]. However, most prior works focus on improving the downstream task performance (*e.g.*, classification), and other advantages of self-supervised learning (*e.g.*, uncertainty or robustness) are rarely investigated [25, 31]. Our work, concurrent with [40, 68], first verifies that contrastive learning is also effective for OOD detection.

Furthermore, we find that the shifting transformations, which were known to be harmful and unused for the standard contrastive learning [5], can help OOD detection. This observation provides new considerations for selecting transformations, *i.e.*, which transformation should be used for positive or negative [66, 71]. Specifically, Tian et al. [66] claims the optimal views (or transformations) of the *positive* pairs should minimize the mutual information while keeping the task-relevant information. It suggests that the shifting transformation may not contain the information for classification, but may contain OOD detection information when used for the *negative* pairs. Xiao et al. [71] suggests a framework that automatically learns whether the transformation should be positive or negative. One could consider incorporating our principle on shifting transformation (*i.e.*, OOD-ness); OOD detection could be another evaluation metric for the learned representations.

## C  Additional one-class OOD detection results

Table 8 presents the confusion matrix of AUROC values of our method on one-class CIFAR-10 datasets, where bold denotes the hard pairs. The results align with the human intuition that 'car' is confused to 'ship' and 'truck', and 'cat' is confused to 'dog'.

Table 9 presents the OOD detection results of various methods on one-class CIFAR-100 (super-class) datasets, for all 20 super-classes. Our method outperforms the prior methods for all classes.

Table 10 presents the OOD detection results of our method on one-class ImageNet-30 dataset, for all 30 classes. Our method consistently performs well for all classes.

Table 8: Confusion matrix of AUROC (%) values of our method on one-class CIFAR-10. The row and column indicates the in-distribution and OOD class, respectively, and the final column indicates the mean value. Bold denotes the values under 80%, which implies the hard pair.

|       | Plane | Car  | Bird | Cat  | Deer | Dog  | Frog | Horse | Ship | Truck | Mean |
|-------|-------|------|------|------|------|------|------|-------|------|-------|------|
| Plane | -     | **74.1** | 95.8 | 98.4 | 94.9 | 98.0 | 96.2 | 90.1  | **79.6** | 82.8 | 90.0 |
| Car   | 99.3  | -    | 99.9 | 99.9 | 99.8 | 99.9 | 99.8 | 99.7  | 98.7 | 95.0  | 99.1 |
| Bird  | 91.1  | 97.5 | -    | 97.3 | 87.0 | 92.5 | 96.1 | 83.2  | 96.4 | 98.0  | 93.2 |
| Cat   | 91.9  | 91.5 | 90.3 | -    | 83.3 | **67.0** | 89.6 | **79.0** | 92.8 | 91.9  | 86.4 |
| Deer  | 95.7  | 98.4 | 94.9 | 96.6 | -    | 94.7 | 98.7 | **69.0** | 97.4 | 98.8  | 93.8 |
| Dog   | 97.9  | 98.5 | 95.5 | 90.3 | 88.1 | -    | 96.8 | **76.6** | 98.6 | 98.3  | 93.4 |
| Frog  | 93.6  | 92.3 | 94.6 | 96.1 | 96.8 | 96.3 | -    | 95.2  | 94.4 | 97.3  | 95.2 |
| Horse | 99.3  | 99.5 | 99.0 | 99.3 | 94.2 | 97.4 | 99.8 | -     | 99.7 | 99.4  | 98.6 |
| Ship  | 96.6  | 91.2 | 99.5 | 99.7 | 99.4 | 99.7 | 99.5 | 99.3  | -    | 96.6  | 97.9 |
| Truck | 96.2  | **72.3** | 99.4 | 99.5 | 99.1 | 99.4 | 98.7 | 98.3  | 96.2 | -     | 95.5 |

Table 9: AUROC (%) values of various OOD detection methods trained on one-class CIFAR-100 (super-class). Each row indicates the results of the selected super-class, and the final row indicates the mean value. * denotes the values from the reference, and bold denotes the best results.

|      | OC-SVM* | DAGMM* | DSEBM* | ADGAN* | Geom* | Rot  | Rot+Trans | GOAD | CSI (ours) |
|------|---------|--------|--------|--------|-------|------|-----------|------|------------|
| 0    | 68.4    | 43.4   | 64.0   | 63.1   | 74.7  | 78.6 | 79.6      | 73.9 | **86.3**   |
| 1    | 63.6    | 49.5   | 47.9   | 64.9   | 68.5  | 73.4 | 73.3      | 69.2 | **84.8**   |
| 2    | 52.0    | 66.1   | 53.7   | 41.3   | 74.0  | 70.1 | 71.3      | 67.6 | **88.9**   |
| 3    | 64.7    | 52.6   | 48.4   | 50.0   | 81.0  | 68.6 | 73.9      | 71.8 | **85.7**   |
| 4    | 58.2    | 56.9   | 59.7   | 40.6   | 78.4  | 78.7 | 79.7      | 72.7 | **93.7**   |
| 5    | 54.9    | 52.4   | 46.6   | 42.8   | 59.1  | 69.7 | 72.6      | 67.0 | **81.9**   |
| 6    | 57.2    | 55.0   | 51.7   | 51.1   | 81.8  | 78.8 | 85.1      | 80.0 | **91.8**   |
| 7    | 62.9    | 52.8   | 54.8   | 55.4   | 65.0  | 62.5 | 66.8      | 59.1 | **83.9**   |
| 8    | 65.6    | 53.2   | 66.7   | 59.2   | 85.5  | 84.2 | 86.0      | 79.5 | **91.6**   |
| 9    | 74.1    | 42.5   | 71.2   | 62.7   | 90.6  | 86.3 | 87.3      | 83.7 | **95.0**   |
| 10   | 84.1    | 52.7   | 78.3   | 79.8   | 87.6  | 87.1 | 88.6      | 84.0 | **94.0**   |
| 11   | 58.0    | 46.4   | 62.7   | 53.7   | 83.9  | 76.2 | 77.1      | 68.7 | **90.1**   |
| 12   | 68.5    | 42.7   | 66.8   | 58.9   | 83.2  | 83.3 | 84.6      | 75.1 | **90.3**   |
| 13   | 64.6    | 45.4   | 52.6   | 57.4   | 58.0  | 60.7 | 62.1      | 56.6 | **81.5**   |
| 14   | 51.2    | 57.2   | 44.0   | 39.4   | 92.1  | 87.1 | 88.0      | 83.8 | **94.4**   |
| 15   | 62.8    | 48.8   | 56.8   | 55.6   | 68.3  | 69.0 | 71.9      | 66.9 | **85.6**   |
| 16   | 66.6    | 54.4   | 63.1   | 63.3   | 73.5  | 71.7 | 75.6      | 67.5 | **83.0**   |
| 17   | 73.7    | 36.4   | 73.0   | 66.7   | 93.8  | 92.2 | 93.5      | 91.6 | **97.5**   |
| 18   | 52.8    | 52.4   | 57.7   | 44.3   | 90.7  | 90.4 | 91.5      | 88.0 | **95.9**   |
| 19   | 58.4    | 50.3   | 55.5   | 53.0   | 85.0  | 86.5 | 88.1      | 82.6 | **95.2**   |
| Mean | 63.1    | 50.6   | 58.8   | 55.2   | 78.7  | 77.7 | 79.8      | 74.5 | **89.6**   |

Table 10: AUROC (%) values of our method trained on one-class ImageNet-30. The first and third row indicates the selected class, and the second and firth row indicates the corresponding results.

| 0    | 1    | 2    | 3    | 4    | 5    | 6    | 7    | 8    | 9    | 10   | 11   | 12   | 13   | 14   |
|------|------|------|------|------|------|------|------|------|------|------|------|------|------|------|
| 85.9 | 99.0 | 99.8 | 90.5 | 95.8 | 99.2 | 96.6 | 83.5 | 92.2 | 84.3 | 99.0 | 94.5 | 97.1 | 87.7 | 96.4 |

| 15   | 16   | 17   | 18   | 19   | 20   | 21   | 22   | 23   | 24   | 25   | 26   | 27   | 28   | 29   |
|------|------|------|------|------|------|------|------|------|------|------|------|------|------|------|
| 84.7 | 99.7 | 75.6 | 95.2 | 73.8 | 94.7 | 95.2 | 99.2 | 98.5 | 82.5 | 89.7 | 82.1 | 97.2 | 82.1 | 97.6 |

# D Ablation study on random augmentation

We verify that ensembling the scores over the random augmentations $\mathcal{T}$ improves OOD detection. However, naïve random sampling from the entire $\mathcal{T}$ is often sample inefficient. We find that choosing a proper subset $\mathcal{T}_{\texttt{control}} \subset \mathcal{T}$ improves the performance for given number of samples. Specifically, we choose $\mathcal{T}_{\texttt{control}}$ as the set of the *most common* samples. For example, the size of the cropping area is sampled from $\mathcal{U}[0.08, 1]$ for uniform distribution $\mathcal{U}$ during training. Since the rare samples, *e.g.*, area near $0.08$ increases the noise, we only use the samples with size $(0.08 + 1)/2 = 0.54$ during inference. Table 11 shows random sampling from the controlled set often gives improvements.

Table 11: AUROC (%) values of our method for different number of random augmentations, under one-class (OC-) CIFAR-10 and CIFAR-100 (super-class). The values are averaged over classes. Random augmentations over the controlled set show the best performance.

| # of samples | Controlled | OC-CIFAR-10 | OC-CIFAR-100 |
|---|---|---|---|
| 4 | - | 92.22 | 87.36 |
| 40 | - | 94.13 | 89.51 |
| 40 | ✓ | **94.31** | **89.55** |

# E Efficient computation of (6) via coreset

One can reduce the computation and memory cost of the contrastive score (6) by selecting a proper subset, *i.e.*, *coreset*, of the training samples. To this end, we run K-means clustering [44] on the normalized features $W_m := z(x_m)/\|z(x_m)\|$ using cosine similarity as a metric. Then, we use the center of each cluster as the coreset. For contrasting shifted instances (4), we choose the coreset for each shifting transformation $S$. Table 12 shows the results for various coreset sizes, given by a ratio from the full training samples. Keeping only a few (*e.g.*, 1%) samples is sufficient.

Table 12: AUROC (%) values of our method for various corset sizes (% of training samples), under one-class (OC-) CIFAR-10, CIFAR-100 (super-class), and ImageNet-30. The values are averaged over classes. Keeping only a few (*e.g.*, 1%) samples shows sufficiently good results.

| Coreset (%) | OC-CIFAR-10 | OC-CIFAR-100 | OC-ImageNet-30 |
|---|---|---|---|
| 1% | 94.22 | 89.27 | 91.06 |
| 10% | 94.30 | 89.46 | 91.51 |
| 100% | 94.31 | 89.55 | 91.63 |

## F    Ablation study on the balancing terms

We study the effects of the balancing terms $\lambda_S^{\text{con}}$, $\lambda_S^{\text{cls}}$ in Section 2.3. To this end, we compare of our final loss (5), without (w/o) and with (w/) the balancing terms $\lambda_S^{\text{con}}$ and $\lambda_S^{\text{cls}}$. When not using the balancing terms, we set $\lambda_S^{\text{con}} = \lambda_S^{\text{cls}} = 1$ for all $S$. We follow the experimental setup of Table 1, *e.g.*, use rotation for the shifting transformation. We run our experiments on CIFAR-10, CIFAR-100 (super-class), and ImageNet-30 datasets. Table 13 shows that the balancing terms gives a consistent improvement. CIFAR-10 do not show much gain since all $\lambda_S^{\text{con}}$ and $\lambda_S^{\text{cls}}$ show similar values; in contrast, CIFAR-100 (super-class) and ImageNet-30 show large gain since they varies much.

Table 13: AUROC (%) values of our method without (w/o) and with (w/) balancing terms, under one-class (OC-) CIFAR-10, CIFAR-100 (super-class), and ImageNet-30. The values are averaged over classes, and bold denotes the best results. Balancing terms give consistent improvements.

|  | OC-CIFAR-10 | OC-CIFAR-100 | OC-ImageNet-30 |
|---|---|---|---|
| CSI (w/o balancing) | 94.28 | 89.00 | 91.04 |
| CSI (w/ balancing) | **94.31** | **89.55** | **91.63** |

## G    Combining multiple shifting transformations

We find that combining multiple shifting transformations: given two transformations $\mathcal{S}_1$ and $\mathcal{S}_2$, use $\mathcal{S}_1 \times \mathcal{S}_2$ as the combined shifting transformation, can give further improvements. Table 14 shows that combining "Noise", "Blur", and "Perm" to "Rotate" gives additional gain. We remark that one can investigate the better combination; we choose rotation for our experiments due to its simplicity.

Table 14: AUROC (%) values of our method under various shifting transformations. Combining "Noise", "Blur", and "Perm" to "Rotate" gives additional gain.

|  | Base | Noise | Blur | Perm | Rotate | Rotate+Noise | Rotate+Blur | Rotate+Perm |
|---|---|---|---|---|---|---|---|---|
| AUROC | 87.89 | 89.29 | 89.15 | 90.68 | 94.31 | **94.65** | **94.66** | **94.60** |

# H    Discussion on the features of the contrastive score (6)

We find that the two features: a) the *cosine similarity* to the nearest training sample in $\{x_m\}$, *i.e.*, $\max_m \text{sim}(z(x_m), z(x))$, and (b) the *feature norm* of the representation, *i.e.*, $\|z(x)\|$, are important features for detecting OOD samples under the SimCLR representation.

In this section, we first demonstrate the properties of the two features under vanilla SimCLR. While we use the vanilla SimCLR to validate they are general properties of SimCLR, we remark that our training scheme (see Section 2.2) further improves the discrimination power of the features. Next, we verify that cosine similarity and feature norm are *complementary*, that combining both features (*i.e.*, $s_{\text{con}}$ (6)) give additional gain. For the latter one, we use our final training loss to match the reported values in prior experiments, but we note that the trend is consistent among the models.

First, we demonstrate the effect of cosine similarity for OOD detection. To this end, we train vanilla SimCLR using CIFAR-10 and CIFAR-100 and in- and out-of-distribution datasets. Since SimCLR attracts the same image with different augmentations, it learns to cluster similar images; hence, it shows good discrimination performance measured by linear evaluation [5]. Figure 3a presents the t-SNE [43] plot of the normalized features that each color denote different class. Even though SimCLR is trained in an unsupervised manner, the samples of the same classes are gathered.

Figure 3b and Figure 3c presents the histogram of the cosine similarities from the nearest training sample (*i.e.*, $\max_m \text{sim}(z(x_m), z(x))$), for training and test datasets, respectively. For the training set, we choose the second nearest sample since the nearest one is itself. One can see that training samples are concentrated, even though contrastive learning pushes the different samples. It complements the results of Figure 3a. For test sets, the in-distribution samples show a similar trend with the training samples. However, the OOD samples are farther from the training samples, which implies that the cosine similarity is an effective feature to detect OOD samples.

(a) t-SNE visualization      (b) Similarities (train)      (c) Similarities (test)

Figure 3: Plots for cosine similarity.

Second, we demonstrate that the feature norm is a discriminative feature for OOD detection. Following the prior setting, we use CIFAR-10 and CIFAR-100 for in- and out-of-distribution datasets, respectively. Figure 4a shows that the discriminative power of feature norm improves as the training epoch increases. We observe that this phenomenon consistently happens over models and settings; the contrastive loss makes the norm of in-distribution samples relatively larger than OOD samples. Figure 4b shows the norm of CIFAR-10 is indeed larger than CIFAR-100, under the final model.

This is somewhat unintuitive since the SimCLR uses the *normalized* features to compute the loss (1). To understand this phenomenon, we visualize the t-SNE [43] plot of the feature space in Figure 4c, randomly choosing 100 images from both datasets. We randomly augment each image for 100 times for better visualization. One can see that in-distribution samples tend to be spread out over the large sphere, while OOD samples are gathered near center.[4] Also, note that the same image with different augmentations are highly clustered, while in-distribution samples are slightly more assembled.[5]

We suspect that increasing the norm may be an *easier* way to maximize cosine similarity between two vectors: instead of directly reducing the feature distance of two augmented samples, one can also increase the overall norm of the features to reduce the *relative* distance of two samples.

| (a) Trend of AUROC | (b) Histogram of norms | (c) t-SNE visualization |

Figure 4: Plots for feature norm.

Finally, we verify that cosine similarity (sim-only) and feature norm (norm-only) are complementary: combining them (sim+norm) gives additional improvements. Here, we use the model trained by our final objective (5), and follow the inference scheme of the main experiments (see Table 7). Table 15 shows AUROC values under sim-only, norm-only, and sim+norm scores. Using only sim or norm already shows good results, but combining them shows the best results.

Table 15: AUROC (%) values for sim-only, norm-only, and sim+norm (*i.e.*, contrastive (6)) scores, under one-class (OC-) CIFAR-10, CIFAR-100 (super-class), and ImageNet-30. The values are averaged over classes. Using both sim and norm features shows the best results.

|  | OC-CIFAR-10 | OC-CIFAR-100 | OC-ImageNet-30 |
|---|---|---|---|
| Sim-only | 90.12 | 86.57 | 83.18 |
| Norm-only | 92.70 | 87.71 | 88.56 |
| Sim+Norm | **93.32** | **88.79** | **89.32** |

# I  Rethinking OOD detection benchmarks

We find that resized LSUN and ImageNet [39], one of the most popular benchmark datasets for OOD detection, are visually far from in-distribution datasets (commonly, CIFAR [33]). Figure 5 shows that resized LSUN and ImageNet contain artificial noises, produced by broken image operations.[6] It is problematic since one can detect such datasets with simple data statistics, without understanding semantics from neural networks. To progress OOD detection research one step further, one needs more *hard* or *semantic* OOD samples that cannot be easily detected by data statistics.

To verify this, we propose a simple detection score that measures the *input smoothness* of an image. Intuitively, noisy images would have a higher variation in input space than natural images. Formally, let $x^{(i,j)}$ be the $i$-th value of the vectorized image $x \in \mathbb{R}^{HWK}$. Here, we define the *neighborhood* $\mathcal{N}$ as the set of spatially connected pairs of pixel indices. Then, the *total variation* distance is given by

$$\text{TV}(x) = \sum_{i,j \in \mathcal{N}} \|x^{(i)} - x^{(j)}\|_2^2. \tag{14}$$

Then, we define the *smoothness score* as the difference of total variation from the training samples:

$$s_{\texttt{smooth}}(x) := |\text{TV}(x) - \frac{1}{M} \sum_m \text{TV}(x_m)|. \tag{15}$$

Table 16 shows that this simple score detects current benchmark datasets surprisingly well.

To address this issue, we construct new benchmark datasets, using a fixed resize operation[7], hence coined LSUN (FIX) and ImageNet (FIX). For LSUN (FIX), we randomly sample 1,000 images from every ten classes of the training set of LSUN. For ImageNet (FIX), we randomly sample 10,000 images from the entire training set of ImageNet-30, excluding "airliner", "ambulance", "parking-meter", and "schooner" classes to avoid overlapping with CIFAR-10.[8] Figure 6 shows that the new datasets are more visually realistic than the former ones (Figure 5). Also, Table 16 shows that the fixed datasets are not detected by the simple data statistics (15). We believe our newly produced datasets would be a stronger benchmark for hard or semantic OOD detection for future researches.

Figure 5: Current benchmark datasets: resized LSUN (left two) and ImageNet (right two).

Figure 6: Proposed datasets: LSUN (FIX) (left two) and ImageNet (FIX) (right two).

Table 16: AUROC (%) values using the smoothness score (15), under unlabeled CIFAR-10. Bold denotes the values over 80%, which implies the dataset is easily detected.

| CIFAR10 → | | | | | | |
|---|---|---|---|---|---|---|
| SVHN | LSUN | ImageNet | LSUN (FIX) | ImageNet (FIX) | CIFAR-100 | Interp. |
| **85.88** | **95.70** | **90.53** | 44.13 | 52.76 | 52.14 | 66.17 |

## J  Additional examples of rotation-invariant images

We provide additional examples of rotation-invariant images (see Table 6 in Section 3.2). Those image commonly appear in real-world scenarios since many practical applications deal with non-natural images, *e.g.*, manufacturing - steel [62] or textile [60] for instance, or aerial [70] images. Figure 7 and Figure 8 visualizes the samples of manufacturing and aerial images, respectively.

Figure 7: Examples of steel (left two) and textile (right two) images.

Figure 8: Examples of aerial images.

## Footnotes

[4]t-SNE plot *does not* tell the true behavior of the original feature space, but it may give some intuition.

[5]We also try the local variance of the norm as a detection score. It also works well, but the norm is better.

[6]It is also reported in https://twitter.com/jaakkolehtinen/status/1258102168176951299.

[7]We use PyTorch `torchvision.transforms.Resize()` operation.

[8]We provide the datasets and data generation code in https://github.com/alinlab/CSI.