[Reviews · NeurIPS 2020]

Review 1

Summary and Contributions: The paper introduces a technique for out-of-distribution (OOD) detection based on recent advances in contrastive self-supervised learning. The idea is to contrast an image against domain-shifted augmentations (mainly, rotations) of the same image. The paper leverages this simple technique in learning a scoring function for detecting OOD samples and shows very good results on standard OOD benchmarks.

Strengths: * The approach is a sound and simple adaptation of the well understood contrastive learning method. * The paper provides extensive empirical results on several OOD benchmarks, with very good numbers.

Weaknesses: * While experimental results seem strong, the current format raises some concerns regarding the fairness of comparison to prior work. Empirical evidence would be stronger if results from [1] are compared to the results of this paper. In addition, results were not directly compared to GOAD [2], and it is not clear the comparison is fair since results in [2] are better than the reported ones. Finally, it's always great to provide standard deviation from multiple runs, which is missing from all tables. * I find the choice of rotation as a distribution shifting augmentation a little unconvincing. While it might work for well curated datasets such CIFAR10 and ImageNet, this approach will not suffice for real datasets. * Several key choices are not well motivated, such as using the norm in the scoring function. References: [1] Hendrycks, Dan, et al. "Using self-supervised learning can improve model robustness and uncertainty." Advances in Neural Information Processing Systems. 2019. [2] Bergman, Liron, and Yedid Hoshen. "Classification-based anomaly detection for general data." arXiv preprint arXiv:2005.02359 (2020).

Correctness: More effort needs to be done in ensuring fairness of comparison to prior methods.

Clarity: The paper is easy to follow. I would prefer including discussion of some key modeling choices in the main text instead of moving it to the appendix.

Relation to Prior Work: Relationship to related work, especially [1] in references above, needs to be more emphasized.

Reproducibility: Yes

Additional Feedback: Post rebuttal ------------------- Thanks for addressing my concerns about experimental results. I would like to apologize for misreading your results and I agree that your comparison with references [1] and [2] are solid. As a result, I have raised my score.


Review 2

Summary and Contributions: The paper presents a simple yet effective method to detect OOD samples: some data transformations, seen as harmful to learn feature representations with contrastive learning, are useful to "augment" the set of OOD samples and can be used as negatives in a contrastive loss. The paper presents a new formulation of the contrastive loss, useful for detecting OOD samples, as shown in the robust experimental section. --- After rebuttal --- I have read the author's response and I maintain my score.

Strengths: The paper is well motivated and the contributions are very clear from the beginning. All choices in the methodology are properly motivated. Moreover, the experimental section presents strong results on different setups, datasets, and the ablation study reveals many insights. The main contribution in this paper can provide key insights for future work along these lines.

Weaknesses: I do no think the paper presents any important weakness, as the scope of the work is clear, methodology seems sound and experiments are robust.

Correctness: The empirical methodology seems correct.

Clarity: The paper is well written and clear.

Relation to Prior Work: The paper does a good job on stating the differences between the presented method and existing techniques. Moreover, it correctly cites the work on top of which authors have build upon and how their work differs on it.

Reproducibility: Yes

Additional Feedback: In l.108-118 it seems as if the only transformation used for positive samples in the contrastive loss is the identity function. Later in l.173-175 it specifies which were the used transformations that are used to generate the positive samples. This was a bit confusing, perhaps l.108-118 should clarify that there is other transformations that, applied to an image, can be considered as positive samples? The statement in l116-119, could use a reference to the specific experiments in which they are based on. In l.207, the authors mention "notable performance in the cases when prior methods often fail" regarding Table 2. What does it mean to fail here? I agree the proposed method is better than the other, but at which threshold are you determining that the other methods fail? Maybe this sentence could be rephrased. In l.241-246, Table 6 is referenced, but it seems that the comments are about Table 5.


Review 3

Summary and Contributions: The paper proposes a contrastive learning-based out-of-distribution detection method by using distributionally-shifted augmentation to construct training example pairs. The main contributions of this paper include a new contrastive learning loss and a novel score function to evaluate the outlierness of OOD samples.

Strengths: 1. The authors found that some augmentations on images can be useful for OOD detection and they proposed the so-called con-SI loss to learn the discriminative representation of in-distribution examples, which is quite novel. 2. Although there is no theoretical guarantee that the proposed score function, it does make sense to distinguish OOD examples from inliers. 3. The experiments are comprehensive, the authors tested the proposed method in various settings and the results are convincible.

Weaknesses: 1. Despite the strengths as mentioned before, the overall novelty is not enough for NeurIPS. The proposed method is highly related to contrastive learning and simCLR, so it can be seen as an incremental research. 2. The proposed method is limited to image data, as the transformation augmentation cannot applied to other types of data, such as temporal data. So I suggest the author add some keywords such as visual or image to the paper title. 3. The paper does not clearly explain why these augmentations are helpful for OOD detection. 4. It is better to report the performance variance in each setting.

Correctness: This paper is technical sound, the formulation and the notations are clear to me and the method is easy to follow.

Clarity: This paper is well-organized and well-written.

Relation to Prior Work: Yes, the authors discussed the differences between their work and major previous works in introduction.

Reproducibility: Yes

Additional Feedback: Two additional questions to the authors: 1. Is the resnet-18 used in this paper pretrained or trained for the task from scratch? 2. As the discrimination scores for OOD examples are various among different classes, how do you set the threshold in practical scenarios.


Review 4

Summary and Contributions: This paper proposes "Contrasting Shifted Instances (CSI)" as an approach of novelty detection. The proposed approach is a simple modification of existing contrastive learning approaches. The main idea of the paper is that in addition to contrasting the samples with other samples, we can contrast samples with augmentations of themselves. This is unlike existing approaches which force augmentations of the same sample to be close to each other. The authors also propose a score which characterizes the novelty of a test sample. The authors conduct extensive experiments to validate the proposed approach.

Strengths: Apart from a few minor short-comings, the paper is very-well written. The authors have clearly and concisely explained their approach. There is extensive experimental validation of the proposed approach. A major strength of the paper is the simplicity of the proposed approach.

Weaknesses: The major weakness of the paper is that the authors do not relate their work with recent work on the same topic. I think a section relating the proposed work to prior work will make the paper much better. In addition, the authors should answer the following questions in the rebuttal/re-submission: 1. The authors have mentioned that con-SI does not improve the representation for standard classification. However, it is not clear whether the proposed approach harms standard classification. The authors should clarify whether the performance for standard classification changes a lot with con-SI. If the performance degrades, by how much? The authors should also discuss if there are ways to avoid such degradations in the performance. 2. Did the authors employ a principled way of selecting the set of transformations S? Currently, it's not clear how did the authors select the number of transformations to use and also which transformations to include in S. They have conducted ablation studies to understand the effectiveness of each transformation separately. However, can the authors discuss more principled ways of searching for elements of S? I think there might be some adversarial methods for selecting the best transformations.

Correctness: Yes. I have verified some of the mathematical formulation in the paper and did not find any major issues.

Clarity: Yes. The authors have clearly explained the approach and experiments.

Relation to Prior Work: Not in depth. This is a major weakness of the paper.

Reproducibility: Yes

Additional Feedback: No changes after the rebuttal. I think this is a good paper.

[Author Response · NeurIPS 2020]

We thank all the reviewers for their valuable comments, efforts, and time. As the reviewers highlight, we propose a simple
(**R1**,**R2**,**R4**) yet effective (**R1**,**R2**) OOD detection method, supported by strong experimental supports (**R1**,**R2**,**R3**,**R4**)
and with a clear presentation (**R1**,**R2**,**R3**,**R4**). We respond to each comment one-by-one in what follows.

———————————————————————— **Common Responses** ————————————————————————

[**R1**/**R4**] **Choice of the shifting transformations.**
Our principle is to choose shifting transformation
that generates the most OOD-like samples. Here,
to measure *OOD-ness* of a transformation, we use

| | (a) OOD-ness | | (b) AUROC: DTD (in) vs. Textile (out) | | |
|---|---|---|---|---|---|
| | Rot. | Noise | SimCLR | CSI (Rot.) | CSI (Noise) |
| | 50.6 | **75.7** | 70.3 | 65.9 | **80.1** |

AUROC between in-distribution vs. transformed samples using vanilla SimCLR (as in line 235-246). Via this concrete
selection scheme, we choose "rotation" (Rot.) for CIFAR/ImageNet (see Table 4). We remark the scheme can be used
for any real-world image datasets. For example, under Describable Texture Dataset (DTD), Rot. is no longer a shifting
transformation (as in line 247-257): the above tables show that (a) "Gaussian noise" (Noise) has higher OOD-ness than
Rot. on DTD, and (b) our method (CSI) significantly benefit from using Noise instead of Rot in this case.

[**R1**/**R3**] **Variance over multiple runs.** As suggested by **R1** and **R3**, we
will include the variance of our results in the final draft, *e.g.*, the right table
presents the mean and standard deviation of the mean AUROC on one-class
CIFAR-10 (averaged over 10 classes) over 5 runs.

| AUROC on one-class CIFAR-10 | | |
|---|---|---|
| Rot+Trans [1] | GOAD [2] | CSI (ours) |
| $89.79_{\pm 0.13}$ | $85.06_{\pm 0.03}$ | $94.27_{\pm 0.05}$ |

[**R1**/**R4**] **Related work.** Following suggestions of **R1** and **R4**, we will add more discussions on related work, particularly
on the connection of our method to the existing self-supervised approaches [1,2], *e.g.*, auxiliary classifier (line 122).

———————————————————————— **Individual Responses** ————————————————————————

[**R1**] **Fairness of comparison.** We put our best efforts to fairly compare our method with the prior works, especially
[1,2], *e.g.*, we already have provided *both* the reported values from [1,2] and results from our re-implementations in
Table 1. The only difference we aware of the experimental details is the use of different architecture, *i.e.*, ResNet-18, for
our method, but we have compensated this issue by using the same architecture in our re-implementations. Somewhat
remarkably, our results on this smaller architecture outperform both results of prior methods with significant margins.

[**R1**] **Motivation on the norm in the scoring function.** Intuitively speaking, the contrastive loss we minimize enforces
the norm to increase, as it is an easier way to maximize cosine similarity (see Appendix I for more details). One of
our key findings is that using the norm of the projection outputs as a score function is a surprisingly strong baseline,
compared to other reasonable choices. We refer more discussions on other design choices in Appendix E, F, and G. For
better presentation, we will add the relevant discussion in the main text following your suggestion.

[**R2**] **Editorial comments.** We will revise our manuscript by clarifying all of the following points: (*i*) line 108-118:
We will remark that Eq. (3) also uses the transformations for positive samples (SimCLR augmentations), which were
implicitly defined in Eq. (2). (*ii*) line 116-118: We will add the navigation: linear evaluation paragraph in Section 3.
(*iii*) line 207: We will rephrase the expression; we stated prior methods often fail, since they show 50% of AUROC (*i.e.*,
random guess) under the Place-365 dataset. (*iv*) line 241-246: We will fix it to Table 5.

[**R3**] **Novelty.** We believe the key novelty of our method belongs to mainly in two aspects: (*i*) We design a surprisingly
effective OOD score functions applicable to any contrastive features, an emerging paradigm for representation learning,
based on an extensive empirical justification. (*ii*) We report a novel observation that some existing input transformation
techniques (*e.g.*, rotation), *i.e.*, *shifting* transformations, could further improve the contrastive features in terms of OOD
detection, under our newly proposed contrastive training scheme. As also highlighted by **R2**, we believe our work could
provide novel insights for both representation learning and OOD detection.

[**R3**] **Limited to the visual modality.** Following the suggestion, we will revise our paper title to emphasize the visual
domain we are focusing on in our experiments. Nevertheless, in principle, our key idea of contrasting transformed
inputs is applicable to other domains, *e.g.*, using Gaussian noise applicable for any data types, including temporal data.

[**R3**] **Why augmentation helps OOD detection?** We claim some augmentations that largely shift data-distribution
would behave like OOD, and contrasting them could help to learn a better discriminative features between in- vs.
out-of-distribution (line 42-50,114-118). This hypothesis is validated by an extensive empirical study (line 235-246).

[**R3**] **Other comments.** (*i*) Pre-train or scratch?: We train all models from scratch (both ours and others). (*ii*) Threshold:
One can set the threshold from training data statistics, controlling the margin for precision/recall trade-off.

[**R4**] **Classification accuracy.** Although classification is not our primary focus, we found that our method achieves
comparable classification accuracy with SimCLR throughout our experiments, *e.g.*, our method achieves test accuracy
of 90.19%, while SimCLR shows 90.48%, under the linear evaluation protocol on CIFAR-10 (see line 260-262).

[1] Hendrycks et al. Using self-supervised learning can improve model robustness and uncertainty. *NeurIPS* 2019.
[2] Bergman and Hoshen. Classification-based anomaly detection for general data. *ICLR* 2020.

[Meta-Review · NeurIPS 2020]

The paper addresses the problem of out of distribution (OOD) detection for computer vision by making sensible changes to the contrastive learning framework. The reviewers agree that the contributions are sufficient for acceptance at NeurIPS, and I am willing to ignore the comment about novelty, given that even though the approach is simple, its effectiveness in the context of OOD detection was now well established. PS: it makes sense for this paper and arxiv.org/abs/2007.05566 to cite each other as concurrent work and discuss the similarities and differences.